# Water Use Efficiency and Stress Tolerance of the Potential Energy Crop *Miscanthus lutarioriparius* Grown on the Loess Plateau of China

**DOI:** 10.3390/plants10030544

**Published:** 2021-03-13

**Authors:** Xuhong Zhao, Lifang Kang, Qian Wang, Cong Lin, Wei Liu, Wenli Chen, Tao Sang, Juan Yan

**Affiliations:** 1Key Laboratory of Plant Resources and Beijing Botanical Garden, Institute of Botany, Chinese Academy of Sciences, Beijing 100093, China; zhaoxh17@ibcas.ac.cn (X.Z.); kanglf@ibcas.ac.cn (L.K.); mm_wang_qian@163.com (Q.W.); lincong@ibcas.ac.cn (C.L.); liuw@ibcas.ac.cn (W.L.); sang@ibcas.ac.cn (T.S.); 2University of Chinese Academy of Sciences, Beijing 100049, China; 3Department of Crop Genomics and Bioinformatics, College of Agronomy and Biotechnology, China Agricultural University, Beijing 100094, China; 4State Key Laboratory of Systematic and Evolutionary Botany, Institute of Botany, Chinese Academy of Sciences, Beijing 100093, China; chenwl@ibcas.ac.cn; 5Key Laboratory of Plant Germplasm Enhancement and Specialty Agriculture, Wuhan Botanical Garden, Chinese Academy of Sciences, Wuhan 430074, China

**Keywords:** *Miscanthus lutarioriparius*, survival rate, water use efficiency, photosynthetic rate, biomass yield

## Abstract

As a potential energy crop with high biomass yield, *Miscanthus lutarioriparius* (*M. lutarioriparius*), endemic to the Long River Range in central China, needs to be investigated for its acclimation to stressful climatic and soil conditions often found on the marginal land. In this study, traits related to acclimation and yield, including survival rates, plant height (PH), stem diameter (SD), tiller number (TN), water use efficiency (WUE), and photosynthetic rates (A), were examined for 41 *M. lutarioriparius* populations that transplanted to the arid and cold Loess Plateau of China. The results showed that the average survival rate of *M. lutarioriparius* populations was only 4.16% over the first winter but the overwinter rate increased to 35.03% after the second winter, suggesting that plants having survived the first winter could have acclaimed to the low temperature. The strikingly high survival rates over the second winter were found to be 95.83% and 80.85%, respectively, for HG18 and HG39 populations. These populations might be especially valuable for the selection of energy crops for such an area. Those individuals surviving for the two consecutive winters showed significantly higher WUE than those measured after the first winter. The high WUE and low stomatal conductance (g_s_) observed in survived individuals could have been responsible for their acclimation to this new and harsh environment. A total of 61 individuals with productive growth traits and strong resistance to cold and drought were identified for further energy crop development. This study showed that the variation of *M. lutarioriparius* held great potential for developing energy crops following continuous field selection.

## 1. Introduction

With the climate change and ecology disruption, attention has been increasingly focused on the development of energy plants capable of adapting to harsh environments such as cold and drought climates [1,2,3,4], which have posed a great challenge to feedstock production [5,6]. It was reported that drought stress reduced plant weight by 45% [7] and cold stress severely limited the survival rate of *Miscanthus* [8]. Therefore, the selection for stress-tolerant plants from genetically divergent populations could provide a valuable solution to this problem [9].

Drought stress is one of the limiting factors to global agricultural production [10,11]. Faced with the intensifying drought, plants may regulate the stoma sizes to reduce water loss but it simultaneously increases the risk of reducing stomatal conductance and photosynthetic rates [12,13]. A study indicated that the maize genotype with drought tolerance had a higher water use efficiency (WUE) when suffering from drought stress [14].

*Miscanthus* has been considered to be a promising second-generation bioenergy crop [15]. The genus is naturally distributed in a wide climatic range from tropical areas of east Asia to ~50 °N in eastern Russia [16,17]. Therefore, *Miscanthus* species have extensive genetic diversity for energy crop development. Currently, much research has focused on the *Miscanthus* × *giganteus,* which is especially suitable for growing in temperate regions [18,19,20]. However, the unfavorable climatic conditions limited the biomass yield of *Miscanthus* × *giganteus* and tolerance to drought and cold due to the single genotype [21,22]. The cultivation of *Miscanthus* species with stress tolerance has been proved an imperative event to ensure high biomass yields in harsh environments [18,23,24,25].

*Miscanthus lutarioriparius* is endemic to the middle and lower reaches of the Yangtze River in China [26], with annual rainfall ranging between 1000 mm and 1400 mm [27]. An early transplanting study was carried out by Yan et al. [8] in the semiarid Qingyang of the Gansu (QG) in China, which has an annual average precipitation of 555 mm [28]. Subsequent studies suggested that *M. lutarioriparius* that was transplanted to the QG had an establishment rate of ~70% and high photosynthetic rates and water use efficiency [29,30]. It was also found that *M. lutarioriparius* had very high genetic variation, especially within populations [26,29,31], which makes the species a promising candidate for energy crop development.

Overwintering rate, growth traits, and photosynthesis-related parameters, especially WUE, are expected to play crucial roles in selecting *M. lutarioriparius* populations and genotypes with cold and drought tolerance. Photosynthetic rate is an important target for crop breeding under stressful growth conditions [32]. Stomatal conductance is highly correlated with stress tolerance [33]. Furthermore, a high WUE can reduce the vulnerability of plants to drought [34]. Instantaneous extrinsic WUE (WUE_e_) is calculated as the ratio of net assimilation rate to transpiration rate (E) in leaf, while intrinsic WUE (WUE_i_) is calculated as the ratio of net assimilation rate to stomatal conductance in leaf [35]. High WUE_e_ contributes to an increase in drought tolerance, while high WUE_i_ is an alternative strategy to improve crop performance under drought conditions [36]. As for C_4_ plants, they can maintain a high photosynthetic rate and productivity even if the low stomatal conductance and transpiration rate occur under drought stress [37,38]. The low transpiration rate for reducing water loss results in a low CO_2_ concentration, which causes a decrease in stomatal conductance and photosynthetic rate [37,39]. The decline in photosynthetic rates and the increase in WUE under drought stress is in favor of plant survival but is unable to contribute to biomass yield [6].

In this study, a total of 41 *M. lutarioriparius* populations were collected from the native distributional regions and transplanted to the arid Huanxian of the Gansu (HG). This study aimed to identify *M. lutarioriparius* individuals or populations with enhanced tolerance to the arid and cold environment of the Loess Plateau. In addition, we hypothesized that successful overwintering might be associated with the changes of growth traits, photosynthetic parameters, and water use efficiency, indicative of a physiological acclimation to the new environment.

## 2. Results

### 2.1. The Variation in the Survival and Overwintering Rates of M. lutarioriparius Populations in Arid and Cold Environments

*M. lutarioriparius* populations had great mortality in the harsh environments in HG. Using the total number of plants grown in 2011, the average survival rate for all populations was 4.16% in 2012. HG41 had the highest survival rate of 27.93% in 2012, while the survival rates of HG26, HG27, HG33, and HG36 populations were less than 0.50%. In the following year, the average survival rate for all populations decreased to 1.45% in 2013. The population with the highest survival rate was HG39, with 9.69% in 2013, followed by populations HG08 and HG41 (Figure 1).

Based on the survival individuals in 2012, the overwintering rates in 2013 were calculated to be 35.03% on average for all populations. Especially, HG18 showed the highest overwintering rate of 95.83% followed by 80.85% of HG39 (Figure 2).

### 2.2. Phenotypic and Physiological Characteristics in 2012 between Survival Individuals and Dead Individuals in 2013

Through the two growing seasons, 634 individuals were survival in 2012, of which 470 individuals were dead in 2013. We divided them into two groups, with 470 dead individuals and 164 survival individuals, and compared their different performances in 2012. We found that the average plant height, stem diameter, and tiller number of survival individuals in 2013 were 161 cm, 7.04 mm, and 24.60 in 2012, respectively. These values were significantly higher than those of dead individuals in 2013 (Figure 3).

By daily dynamic curves, we found the photosynthetic rate was maintained above 30 µmol m^−2^ s^−1^ from 10:00 a.m. to 17:00 p.m., and the highest *A* occurred at 13:00 p.m. (Appendix A). Corresponding with phenotypic traits, we found the average photosynthetic rate and WUE_e_ of the two groups were 32.25 μmol m^−2^ s^−1^ and 31.22 μmol m^−2^ s^−1^, and 7.11 mmol mol^−1^ and 6.87 mmol mol^−1^ in 2012, respectively. There were no significant differences in photosynthetic rate and WUE_e_ between them. The average stomatal conductance, intercellular CO_2_ concentration, transpiration rate in 2012 presented a significantly higher value in those 470 individuals. Meanwhile, a significantly higher WUE_i_ (*p* < 0.01) was observed in 2012 in 164 individuals (Figure 3).

### 2.3. The Phenotypic and Physiological Changes of M. lutarioriparius Survived in Two Growing Seasons

A total of 23 populations and 155 surviving individuals were characterized for the phenotypic and physiological changes in the two growing seasons. The average plant heights were 159.80 cm in 2012 and 153.97 cm in 2013, which was no significant difference between the two growing seasons. However, the average tiller number significantly (*p* < 0.01) reduced from 23.07 in 2012 to 5.57 in 2013. The average photosynthetic rate decreased from 29.93 μmol m^−2^ s^−1^ in 2012 to 27.51 μmol m^−2^ s^−1^ in 2013, the transpiration rate from 4.54 mmol m^−2^ s^−1^ in 2012 to 2.62 mmol m^−2^ s^−1^ in 2013, and the WUE_i_ from 164.33 μmol mol^−1^ in 2012 to 141.22 μmol mol^−1^ in 2013. The reduction presented a greatly significant level. However, the average stomatal conductance, intercellular CO_2_ concentration, and WUE_e_ were significantly increased from 2012 to 2013, respectively (Figure 4).

Two-way ANOVA was conducted for 23 populations to study the effect of population and growing season on the growth traits, photosynthetic parameters, and water use efficiency. The results showed that different populations had a significant effect on all growth traits, photosynthetic parameters, and water use efficiency (Table 1). Moreover, a significant interaction between growing seasons and populations was found on stem diameter, tiller number, photosynthetic rate, transpiration rate, WUE_e_, and WUE_i_.

### 2.4. Correlation Properties and Individual Classifications of Surviving M. Lutarioriparius in Arid and Cold Environments

Correlation analyses were conducted on growth traits, photosynthetic parameters, and water use efficiency of 155 surviving individuals in 2013. The results indicated that photosynthetic rate, transpiration rate, and WUE_i_ were significantly positively correlated with tiller number, whereas a negative correlation was observed between tiller number and intercellular CO_2_ concentration and WUE_e_. In addition, photosynthetic rate, intercellular CO_2_ concentration, and transpiration rate were positively correlated to stomatal conductance (Table 2).

Cluster analysis and heatmap were performed on 155 survived individuals based on growth traits, photosynthetic parameters, and water use efficiency for studying the features of growth and photosynthesis parameters. The results indicated that 155 individuals were clustered into 5 groups (Figure 5). Those individuals in the first group (red lines including 36 individuals) possessed a large tiller number and a high photosynthetic rate, stomatal conductance, intercellular CO_2_ concentration, and transpiration rate, but the value of plant height, stem diameter, WUE_e_, and WUE_i_ only displayed at the low or middle level. The second group of *M. lutarioriparius* (blue lines), including 41 individuals, showed a high plant height and thick stem diameter, while the value of tiller number, photosynthetic parameters, and water use efficiency reached a middle level. Individuals in the third group of *M. lutarioriparius* (black lines including 41 individuals) showed the lowest value in three growth traits and photosynthetic parameters and water use efficiency presented at the low or middle level. The fourth group of *M. lutarioriparius* (green lines), including 20 individuals, displayed a high plant height, thick stem diameter, and high WUE_i_ but a low value of photosynthetic rate, stomatal conductance, intercellular CO_2_ concentration, transpiration rate, and middle-level WUE_e_. The last group of *M. lutarioriparius* included 17 individuals, which showed a low plant height, thin stem diameter, and relatively low values of photosynthetic parameters but a large tiller number and higher WUE_i_.

## 3. Discussion

Whether plants could successfully establish in a new harsh environment depends on many factors, such as annual precipitation, temperature, soil type, and management patterns [40]. We found the total survival rate for all *M. lutarioriparius* populations was only 4.16% in 2012, which was much lower than the establishment and overwintering rate of *M. lutarioriparius* in Qingyang of the Gansu province (QG) [8]. Because the experiment in HG aimed to identify strong tolerant individuals or populations by natural selection, there was little watering or management in the early planting time. Additionally, the annual precipitation was 369.9 mm in 2011 in HG and 459.6 mm in 2009 in QG. The temperature was also colder in HG than in QG. These reasons might impact seriously the survival rate of seedlings. Selecting *M. lutarioriparius* with drought and cold tolerance could become an important step for improving establishment. The large variation of survival rate between populations could benefit the development of *M. lutarioriparius* with cold and drought tolerance. For example, population HG39 had a survival rate of 11.99% and 9.69% in 2012 and 2013, respectively. The 9.69% was the highest survival rate in all populations. It means that survivors in population HG39 may have the cultivating potential in the arid and cold regions.

The previous study indicated that the low temperature limited the distributes of C_4_ plant in the Loess Plateau [41]. It implied that low temperature was a vital factor affecting the survival of C_4_ plants. *Miscanthus*, a promising C_4_ bioenergy crop, exhibited an overwintering loss of 29% and the survivors still suffered severe damage from cold temperature [18]. Yan et al. [8] transplanted 14 *M. lutarioriparius* populations to cold Xilinhot of the Neimenggu Autonomous Region for the domestication experiment, but only two populations survived. Peixoto et al. [19] suggested that −6.5 °C temperature would make *Miscanthus* rhizomes lose the reproductive ability. Although the stronger cold tolerance of *M. sacchariflorus* than *M. lutarioriparius* had been reported in a common garden experiment located in northern China [8], −3.4 °C temperature could still kill 50% *M. sacchariflorus* individuals [42]. In the present study, the extremely low ground temperature from 2011 to 2012 in HG reached −18.6 °C which could have caused the death of most *M. lutarioriparius* individuals. The cold temperature in winter had been a key factor limiting the survival of *M. lutarioriparius* populations in the Loess Plateau of China.

However, we found that the overwintering rate of some *M. lutarioriparius* populations survived over the first winter was more than 50% after experiencing a cold winter with extreme ground temperature −21.9 °C from 2012 to 2013. For example, the overwintering rate of population HG07, HG09, HG10, HG13, and HG21 was more than 50% while population HG39 and HG18 reached 80.85% and 95.83% in 2013. Furthermore, certain individuals not only survived in cold winter but also showed a productive property with plant height more than or close to 3 m, including individual HN07-23, HN21-20, HN05-8, HN10-7, and HN09-18. These populations and individuals were expected to be useful for breeding productive and cold-tolerant energy crops in HG.

Moreover, our study showed that those individuals that survived into 2013 had significantly greater plant height, thicker stem diameter, and larger tiller number in 2012 than those that did not. It indicated that poor performance in growth traits may be an important factor that contributed to the low survival rate. However, the reasons that short plant height, small stem diameter, and less tiller caused the low survival rate needed further study. Those individuals that survived into 2013 displayed a significantly higher WUE_i_ in 2012 than those that did not. High WUE was beneficial to the survival of individuals in arid environments [6,29]. This is in agreement with our findings that high WUE_i_ might be an important trait related to the high survival rate. A recent study indicated that the increase in the WUE_i_ could enhance plant growth [43]. In our study, WUE_i_ had a significantly positive correlation with tiller number. It indicated that a larger tiller number was coupled with higher WUE_i_, which might be beneficial to higher productivity.

It was reported that *M. lutarioriparius* was able to maintain remarkably high photosynthetic rates when transplanted to a colder and drier location [30]. Our study showed that *M. lutarioriparius* populations had an average photosynthetic rate of 30.90 μmol m^−2^ s^−1^ in August 2012 in drought HG, which was much higher than that of the 27.10 μmol m^−2^ s^−1^ measured in the same growing month in semiarid marginal land [30]. C_4_ plants capable of maintaining a high photosynthetic rate and productivity even if the drought stress had caused a reduction in stomatal conductance and transpiration [37,38]. Our study found that individuals that survived in 2013 and those that did not have similar photosynthetic rates in 2012, but the average stomatal conductance, transpiration rate, and intercellular CO_2_ concentration of the former were significantly lower than the latter. Reduced stomatal conductance could usually cope with the stressful environment [44]. The lower stomatal conductance of those individuals that survived in 2013 might have had improved tolerance to drought.

Plants under stress often presented many adaptable changes in phenotype and physiology [13,45,46]. Our study showed that the tiller number of *M. lutarioriparius* reduced significantly from 2012 to 2013, but there were no significant changes in plant height or stem diameter. The reduction of tiller number was also observed in the previous transplanting study [9]. Thus, the tiller number of *M. lutarioriparius* might be the trait that most easily undergo an adaptive change in drought and cold environment. There was a significant reduction in photosynthetic rate, transpiration rate, WUE_i*,*_ and growing traits in surviving individuals from 2012 to 2013, which might be a strategy for resource allocation [47]. For example, most of *Populus × euramericana* genotypes with productive properties showed a weak ability to drought tolerance, while the genotypes with low productivity presented a wide range of drought tolerance [48]. The WUE_e_ was significantly increased from 2012 to 2013 in the present study. High WUE_e_ was beneficial to the survival of individuals in arid environments [7,29]. The increase in WUE_e_ seemed to be an adaptive change in the drought environment [49]. This indicated that *M. lutarioriparius* could acclimate to the drought environment by improving WUE_e_. Studies of wheat indicated that there was a negative correlation between leaf size and water use efficiency [50] and wheat with short tillers and small leaves tended to have higher photosynthetic rates [51,52]. Therefore, high productivity and strong tolerance to drought need to be balanced in the breeding process of *M. lutarioriparius*.

Therefore, the studies of productivity and stress tolerance are of great importance for energy crop development. We divided *M. lutarioripurius* individuals that survived cold winters and drought seasons into five categories by the growth traits, photosynthetic parameters, and water use efficiency. Those categories with high plant height, thick stem diameter, more tiller number, higher photosynthetic rate, and high WUE_i_ and WUE_e_ should be genotypes with breeding potential. As such, the second and fourth groups with 61 individuals are most likely to contain the candidates.

## 4. Materials and Methods

### 4.1. The Basic Situations of the Experimental Site

The experimental site was located in Huanxian of the Gansu on the Loess Plateau (HG: N 36.02°–37.15°, E 106.35°–107.75°, altitude 1100–2080 m) (Figure 6). The weather data of the experimental site were collected from the China meteorological data network (http://data.cma.cn, accessed on 1 January 2011 to 31 December 2013). The lowest air temperature in the winter from 2011 to 2012 was −20.9 °C, while it was −19.2 °C in the winter from 2012 to 2013 in HG. At the same time, the lowest ground temperature in the winter from 2011 to 2012 was −18.6 °C, while it was −21.9 °C in the winter from 2012 to 2013 (Appendix A). The daily maximum air temperature was below 0 °C for 28 days from November 2011 to May 2012 and 20 days from November 2012 to May 2013, respectively. Meanwhile, the daily maximum ground temperature was below 0 °C for 8 days from November 2011 to May 2012 and 3 days from November 2012 to May 2013 (Appendix A). Annual precipitation was 369.9 mm in 2011, 540.5 mm in 2012, and 674.5 mm in 2013 in HG. Cumulative monthly precipitations in August 2012 and 2013 were 96.2 mm and 59.7 mm, respectively (Appendix A). Cumulative precipitations before the measurement of photosynthetic parameters in 2012 and 2013 were shown in Appendix A**.**

### 4.2. Plant Materials and Experiment Design

A total of 41 *M. lutarioriparius* populations were transplanted to the experimental site from the native habitats of the various geographical locations along the middle and lower reaches of the Yangtze River (Figure 6, Appendix A). The geographical traits of native habitats were greatly different from the experimental site, such as the average altitude of 5–100 m, the annual average temperature of 14–18 °C, and annual precipitation with 1000–1400 mm [27]. First, the seeds of 41 *M. lutarioriparius* populations were cultivated individually to the seedlings in the greenhouse in April 2011. Then, these seedlings were transplanted, respectively, to the experimental field in HG in June 2011 when the plant height grew to 30–40 cm. Forty-one plots were designed as 20 m × 20 m size in the experiment field. Lastly, a total of 32,144 seedlings were transplanted to the experimental field in June 2011. Each individual was with an interval of 0.7 m to avoid plants mixing. Within the first 7 days, we watered these seedlings interval one day to ensure the survival rate of transplanting. After the first 7 days, we stopped to water. The seedlings grew up naturally without management.

During the two growing seasons, the survival rates and overwintering rates of 41 *M. lutarioriparius* populations were calculated, respectively. The survival rate was the ratio of the surviving individuals in June 2012 or 2013 to planting individuals in 2011. The overwintering rate was the ratio of surviving individuals in June 2013 to surviving individuals in June 2012. Additionally, we labeled the surviving individuals in June 2012 and made them as the further studying materials. In August 2012, we measured their growth-related traits, including plant height, stem diameter and tiller number, and photosynthesis-related parameters, including photosynthetic rate, stomatal conductance, transpiration rate, and intercellular CO_2_ concentration. In August 2013, we again measured the same parameters of those surviving and labeled individuals, respectively. There were 468 individuals who survived in a cold winter from 2012 to 2013. Of those surviving individuals in 2013, 164 individuals had been labeled in 2012. We deleted the populations with fewer than or equal to 5 individuals in 2013, 155 individuals from 23 populations were used to estimate the acclimation in growth, photosynthetic parameters, and water use efficiency to harsh environments in 2012 and 2013.

### 4.3. Growth Traits Measurement

Growth traits of each individual, including plant height, stem diameter, and tiller number, were measured, respectively, on 20 and 21 August of 2012 and 9 and 10 August of 2013. The plant height was measured from the base to the highest point of the plant. Stem diameter was measured from the largest tiller at 5 cm above the base by the digital caliper. Tiller numbers were counted within one square meter for each individual.

### 4.4. Photosynthetic Parameters Measurement

Photosynthesis-related parameters, including photosynthetic rate, stomatal conductance, intercellular CO_2_ concentration, and transpiration rate, were logged by using the LI-6400 portable photosynthesis system (LI-COR 6400 XT system; LI-COR, Lincoln, NE, USA) with a standard 6 cm^2^ cuvette. Before conducting the measurements, the daily dynamics of photosynthetic rate were firstly recorded with one time every hour from 7:00 to 19:00 on 21 August 2012 for identifying the daily photosynthetic patterns of *M. lutarioriparius* in natural conditions in HG. Subsequently, the measuring processes of the photosynthesis-related parameters were performed from 10:00 a.m. to 12:00 noon on a clear day on 22 and 23 August 2012 and 12 and 14 August 2013 in the HG, respectively. Measurements were performed on the middle part of the fourth fully expanded leaf from the top to the bottom, under ambient temperature and photon flux density, while CO_2_ reference concentration was maintained at 400 µmol mol^−1^. An infra-red gas analyzer (IRGA, LI-COR, Lincoln, NE, USA) was used to reach equilibrium (monitor ΔCO_2_ and ΔH_2_O) every 20 min. The photosynthesis-related parameters were logged three times for each plant. Instantaneous extrinsic water use efficiency and intrinsic water use efficiency were, respectively, calculated as the ratio of photosynthetic rate to transpiration rate (*A*/*E*) and the ratio of photosynthetic rate to stomatal conductance (*A*/*g_s_*) [30,53].

### 4.5. Data Analysis

The daily dynamics of photosynthetic rate were drawn by origin 9.0 (Origin Lab, Northampton, MA, USA). The survival rates and overwintering rates of *M. lutarioriparius* populations; average plant height, stem diameter, and tiller number; and photosynthetic rate, stomatal conductance, transpiration rate, intercellular CO_2_ concentration, and water use efficiency of surviving individuals were presented in the bar plots by origin 9.0. One-way analysis of variance (ANOVA) was used to compare the difference in the growth-related traits, photosynthesis-related parameters, and water use efficiency in 2012 between those surviving individuals in 2013 and those that died in 2013 by R (version 3.6.2, R Core Team, 2019, Vienna, Austria) The University of Auckland, Auckland, New Zealand). Two-way ANOVA was conducted on the growth-related traits, photosynthesis-related parameters, and water use efficiency of surviving populations between growing years by R for studying the effect of populations and growing years and their interaction. Pearson correlation analysis was conducted on growth traits and photosynthesis-related parameters of surviving individuals in 2013 by R. In order to screen individual groups with high productivity and stress tolerance characteristics, cluster analysis with heatmap was conducted on individuals who survived in 2013 by method “ward.D2” in R (version 3.6.2, R Core Team) based on the ”euclidean” between individuals.

## 5. Conclusions

Our study showed that *M. lutarioriparius* was capable of acclimating to the stressful environment by going through a series of phenotypic and physiologic changes. Although the average survival rate of transplanted populations was lower than 5% over the first winter that was considerably colder than that in their native habitats, the survival rate raised to over 35% over the following winter. In particular, the highest survival rates over the second winter reached above 95% for population HG18. Greater plant height, thicker stems, and higher tiller numbers together with higher water use efficiencies and lower stomatal conductance might have contributed to the cold and drought tolerance. Considering the combination of these traits, sixty-one *M. lutarioriparius* individuals surviving the two consecutive winters were identified as candidates for energy crop development in the Loess Plateau of China. The study also suggested that this outcrossing species possessed a great level of genetic variation that could allow the selection of traits beneficial for both establishment and biomass production of the energy crop under the unfavorable climatic conditions often presented on the marginal land.

## Figures and Tables

**Figure 1 plants-10-00544-f001:**
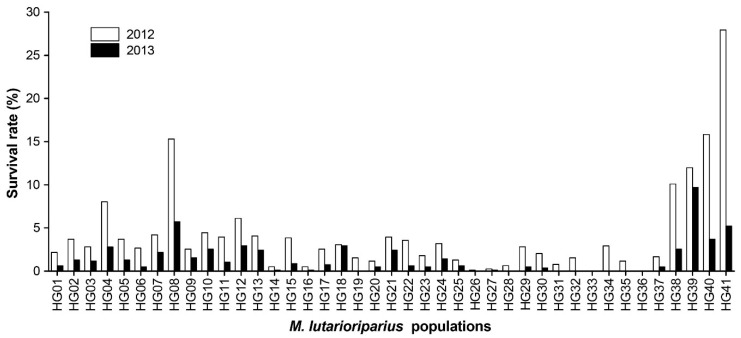
The variation in the survival rate of *Miscanthus lutarioriparius* populations in two growing seasons. The black bars represent the survival rates of populations in the 2012 growing season and the grey bars represent the survival rates of populations in the 2013 growing season, all based on the total number of plants grown in 2011.

**Figure 2 plants-10-00544-f002:**
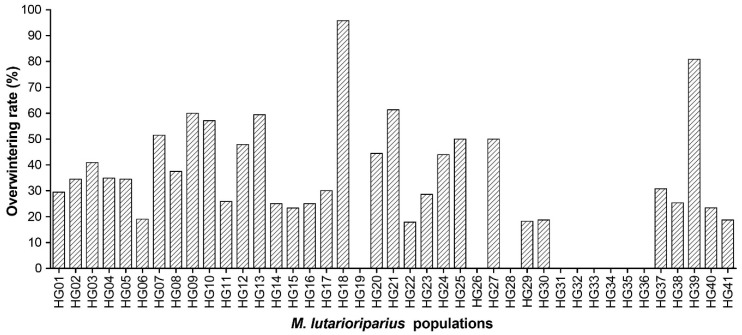
Overwintering rate of 41 *M. lutarioriparius* populations based on surviving individuals in 2012 and 2013.

**Figure 3 plants-10-00544-f003:**
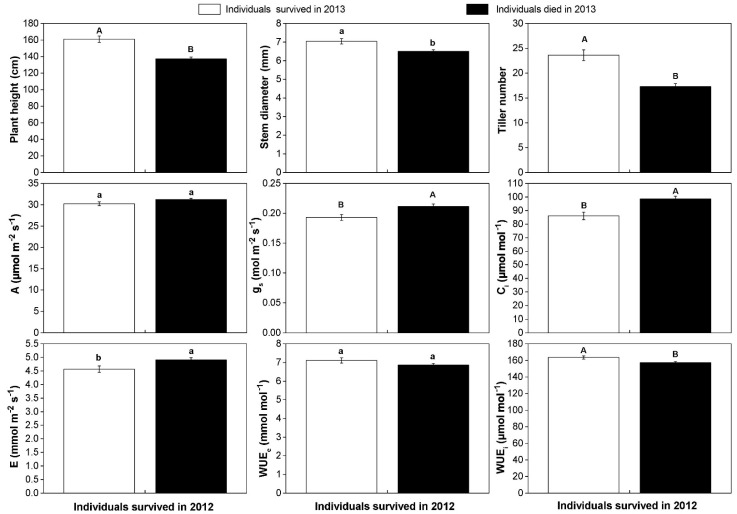
The differences in growth traits, photosynthetic parameters, and water use efficiency of the two groups in 2012. The white bars represented 164 individuals that survived in both 2012 and 2013 and the black bars represented 470 individuals that died in 2013. Mean values and ±SE (error bars) were calculated, respectively, within surviving individuals in 2012. Different lowercases present a significant difference at *p* < 0.05 level (a, b) and capital letters present a significant difference at *p* < 0.01 level (A, B). A: photosynthetic rate; E: transpiration rate; g_s_: stomatal conductance; C_i_: intercellular CO_2_ concentration; WUE_e_: extrinsic water use efficiency; WUE_i_: intrinsic water use efficiency.

**Figure 4 plants-10-00544-f004:**
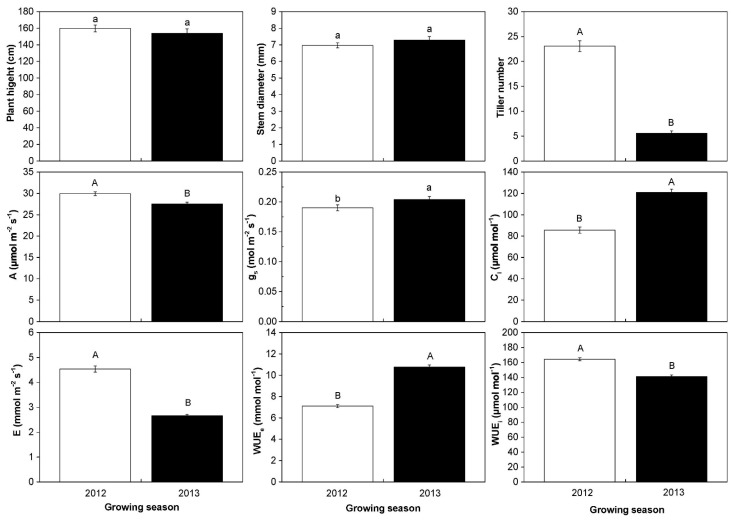
The changes in growth traits, photosynthetic parameters, and water use efficiency when 155 *M. lutarioriparius* individuals acclimated to the harsh environments from 2012 to 2013. The white bars represent the 2012 growing season (2012) and the black bars represent the 2013 growing season (2013). Mean values and ±SE (error bars) were calculated within 155 surviving individuals. Different lowercases present a significant difference at *p* < 0.05 level (a, b) and different capital letters present a significant difference at *p* < 0.01 level (A, B). A: photosynthetic rate; E: transpiration rate; g_s_: stomatal conductance; C*_i_*: intercellular CO_2_ concentration; WUE_e_: extrinsic water use efficiency; WUE_i_: intrinsic water use efficiency.

**Figure 5 plants-10-00544-f005:**
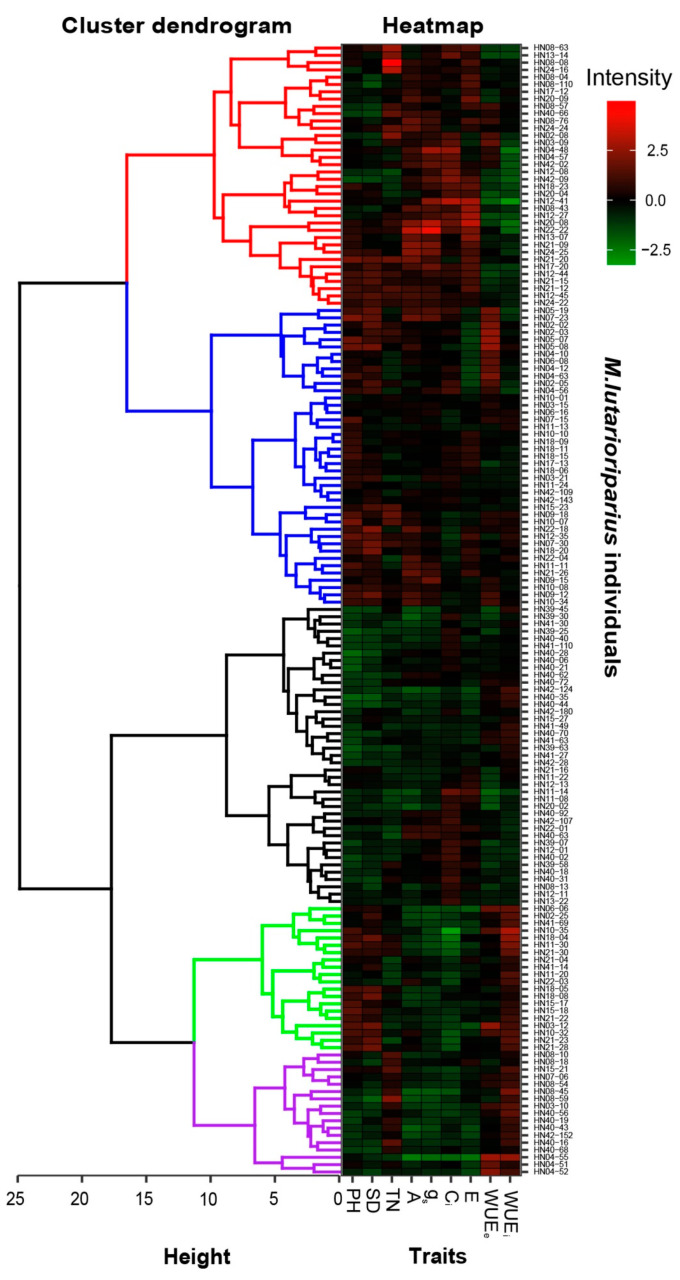
Cluster analysis of 155 individuals survived in both 2012 and 2013. The tree structure on the left presents the 5 groups of individuals and the colored squares on the right present the standardized data of growth traits, photosynthetic parameters, and WUE. The changes in the color in a square from green to red represent the value change of growth traits, photosynthetic parameters, and WUE from low to high.

**Figure 6 plants-10-00544-f006:**
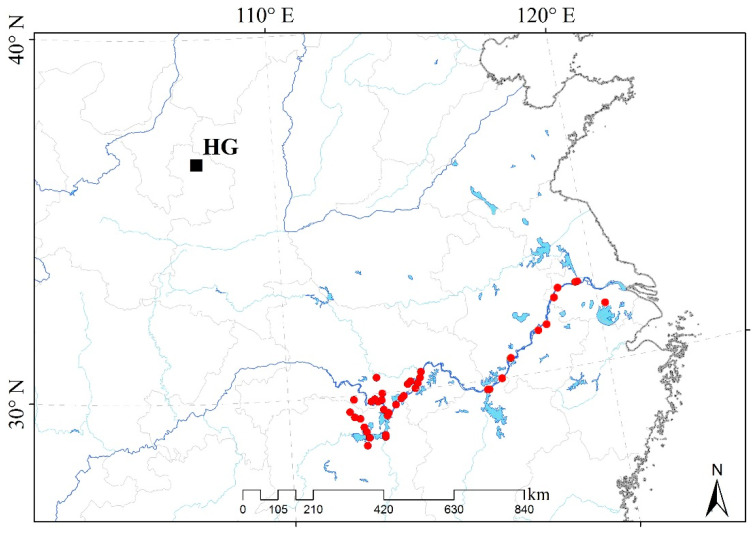
The locations of *M. lutarioriparius* in native habitats and domestication sites in the HG. The black square represents the experimental field in this study; the red dots represent the collection locations of 41 *M. lutarioriparius* populations.

**Table 1 plants-10-00544-t001:** Two-way ANOVA on growth traits and photosynthetic parameters of 155 *M. lutarioriparius* individuals in the arid Huanxian of the Gansu (HG). Year: growing seasons; Pop: populations. PH: plant height; SD: stem diameter; TN: tiller number; A: photosynthetic rate; E: transpiration rate; g_s_: stomatal conductance; C_i_: intercellular CO_2_ concentration; WUE_e_: extrinsic water use efficiency; WUE_i_: intrinsic water use efficiency. MS: Mean square

Traits	Source of Variation	df	MS	*F*	*p*
PH	Year	1	1912	1.19	0.28
	Pop	22	24,766	15.37	<0.001
	Year × Pop	22	1862	1.16	0.29
SD	Year	1	9.01	2.77	0.1
	Pop	22	28.99	8.92	<0.001
	Year × Pop	22	6.55	2.01	0.01
TN	Year	1	24,591	261.13	<0.001
	Pop	22	324	3.42	<0.001
	Year × Pop	22	164	1.74	0.02
A	Year	1	278.1	11.89	0.001
	Pop	22	74.88	3.2	<0.001
	Year × Pop	22	56.23	2.4	0.001
g_s_	Year	1	0.02	4.48	0.035
	Pop	22	0.01	1.88	0.011
	Year × Pop	22	0.01	1.54	0.06
C_i_	Year	1	79,849.47	64.85	<0.001
	Pop	22	2722.37	2.21	0.002
	Year × Pop	22	1671.63	1.36	0.135
E	Year	1	199.27	280.58	<0.001
	Pop	22	5.83	8.21	<0.001
	Year × Pop	22	5.38	7.57	<0.001
WUE_e_	Year	1	833.34	562.32	<0.001
	Pop	22	26.33	17.77	<0.001
	Year × Pop	22	23.15	15.62	<0.001
WUE_i_	Year	1	32,015.11	55.42	<0.001
	Pop	22	956.74	1.66	0.035
	Year × Pop	22	1053.36	1.82	0.015

**Table 2 plants-10-00544-t002:** The correlation analysis between growth traits and photosynthetic parameters based on the 2013 growing season. ** *p* < 0.01, * *p* < 0.05. PH: plant height; SD: stem diameter; TN: tiller number; A: photosynthetic rate; E: transpiration rate; g_s_: stomatal conductance; C_i_: intercellular CO_2_ concentration; WUE_e_: extrinsic water use efficiency; WUE_i_: intrinsic water use efficiency.

Traits	PH	SD	TN	A	g_s_	C_i_	E	WUE_e_
SD	0.74 **							
TN	0.11	0.01						
A	0.29 **	0.23 **	0.24 **					
g_s_	0.15 **	0.13 *	−0.01	0.80 **				
C_i_	−0.20 **	−0.12 *	−0.30 **	0.19 **	0.69 **			
E	0.09	−0.07	0.49 **	0.52 **	0.39 **	0.05		
WUE_e_	0.14 *	0.24 **	−0.44 **	−0.06	−0.02	0.03	−0.80 **	
WUE_i_	0.06	0.02	0.26 **	−0.35 **	−0.80 **	−0.97 **	−0.14 *	−0.01

## Data Availability

The data are available from the corresponding author upon reasonable request.

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
