# Peer review of "Water Use Efficiency and Stress Tolerance of the Potential Energy Crop Miscanthus lutarioriparius Grown on the Loess Plateau of China"

_plants, 2021, doi:10.3390/plants10030544_

Round 1
Reviewer 1 Report
Review of Zhao et al.: „Water use efficiency and stress tolerance of the potential energy crop Miscanthus lutarioriparius grown on the Loess Plateau of China”; resubmitted to Plants, January 2021
I appreciate the changes that have been made to the manuscript including all the additional information that I had asked for in the previous round and that has been added to the supplements. I also think that already added to the quality of the manuscript. But one major concern still is the language and grammar. This needs to be considerably improved. Please take care to do so. Of course, I still have a list of comments that also needed to be addressed (see below). I hope you find them useful.
Introduction:
Overall, the introduction provides all the information needed and also the motivation for and aims of the study are presented relatively clear with some minor exceptions (see comment 1. below).
- In ll.134: It is stated that individuals that successfully overwintered might undergo adaptive changes. However, adaptation is the product of genetic variation and natural selection and occurs at the population level, hence from an evolutionary point of view, an individual cannot undergo adaptive changes. In my opinion, you might rather avoid the term adaptive here and also the term “adaptability” in the previous sentence and rephrase the hypotheses/aims, e.g.:
“The aim of this study was to identify M. lutarioriparius individuals (or populations) with enhanced tolerance to the arid and cold environment of the Loess Plateau. In addition, we hypothesised that successful overwintering might be associated with changes of growth traits, photosynthetic parameters and water use efficiency, indicative of a physiological acclimation to the new environment.”
- 138: The introduction ends with: “[…]water use efficiency, etc…” Please change this.
Results
- 152: in that sentence (starting with: “The highest survival rate of the population was 9.69% in 2013[…]”) HG39 appears two times, once in the middle of the sentence and once at the end. Please check!
- A general remark is that I would wish for one or two introductory sentences at the beginning of each results section and also a conclusive sentence harbouring the main conclusion from that respective section. I think that would greatly enhance the readability of the manuscript. For example, for the first section 2.1:
“In order to identify M. lutarioriparius populations with enhanced tolerance to cold and arid climates individuals from 41 populations were grown in the Loess Plateau and their survival rates were monitored across two consecutive growing seasons in 2012 and 2013.” Then follow your results on the survival rates.
Then you might introduce the overwintering rates with another brief sentence: “Based on the individuals that survived the first year of the experiment overwintering rates for the second year were calculated. The average overwintering rate…”
But this really applies to all sections in the results.
- Figure 1: To me it is not clear which panel of figure 1 is to be kept in the final version of the manuscript but I suppose it is the lower one. Therefore, please correct the figure caption, which states that there are black and grey stripes. Also, it should read bars instead of stripes.
- Figure 2: Some of the populations are denoted with “HNxx” instead of “HGxx”. Please correct. This also applies to Figure 1, Figure 6, your supplementary tables and parts of the discussion.
- Again, please provide a clear statement at the beginning of section 2.2 of what you have been doing and why, e.g.:
“In order to assess whether successful overwintering was preceded by detectable differences in plant performance, several growth-related parameters were measured in August 2012. In 2013, following the second winter, the previously assessed plants were assigned to two groups based on whether they successfully overwintered or not, i.e. individuals that survived in 2013 and individuals that died in 2013, respectively. The results…”.
“In addition to growth-related traits, several photosynthesis-related parameters were also assessed. All of these traits were measured between 10:00 hours and 12:00 hours since preliminary measurements of the daily dynamic curves indicated that the photosynthetic rate was maintained above 30 µmol m-2 s-1 between 10:00 hours and 17:00 hours with the highest value occurring at 13:00 hours (Figure S1). The results…”
- In the caption of figure 3 could you please mention what A, E, gs and ci denote? Also, in the caption you say that the white bars represent one population that survived in both 2012 and 2013, and the black bars represent the other population including 470 individuals that died in 2013. As far as I understood you simply grouped all individuals together that either survived or died, irrespectively of the populations they belong to. If this is the case, please avoid using the term population here since this is misleading. Please also indicate the statistical test that has been used to analyse the data.
- Section 2.3: Could you please state here why you ended up with only 155 individuals instead of the 164 mentioned in the previous sections? Please also briefly state how many of your populations were represented within these 155 individuals. Please also indicate what the different parameter abbreviations stand for and which statistical test you have used for the analysis.
- Thanks for including the table with the ANOVA results. Could you please either indicate the residual degrees of freedom, or, alternatively, mention the number of individual plants that have been considered for these analyses? Please also try to have a better separation between the different traits, e.g., by using bigger line spacing between the traits or by using a separating line.
- In ll.259 please be clearer about which traits correlate to each other, e.g.: “Photosynthetic rate, transpiration rate and WUEi were significantly positively correlated to tiller number, whereas a negative correlation was observed between tiller number and intercellular CO2 concentration and WUEe, respectively. In addition, photosynthetic rate, intercellular CO2 concentration and transpiration rate were all significantly positively correlated to stomatal conductance.”
- What do you conclude from the cluster analysis besides the fact that certain individuals from clusters 2 and 4 might be especially suited for breeding cold-/drought-tolerant M. lutarioriparius genotypes? Wouldn’t you expect that individuals from the same population should cluster together? At least if you assume that the intra-specific variation that you observed is due to genetic variation between populations? However, in your case for many populations the individuals are distributed across four or five of the different clustering groups. Could you please discuss this in more detail in the discussion? What is known about the genetic variability in this species? In the introduction, at least, you briefly mention two publication from your own group showing that there was “abundant within populations and among populations genetic variation” (ll.104).
Discussion
- The second paragraph of the discussion starts with a sentence I do not understand. Please rewrite.
- Sometimes populations are denoted “HNxx” and sometimes “HGxx”. Please use uniform nomenclature for your populations (see comment further above).
- 337: “[…]and so on.” Please remove this or write clearly what you mean.
- In ll.369 you write about how drought would reduce tiller number and cite a reference (Kunkhonnuruk et al. 2010 – Industrial Crop & Products). However, that reference is about something completely different. Please check all your references and make sure they are correct.
- As stated above, I miss information about Miscanthus genomics/genetic variability.
- I am fine with the conclusion that you might have identified certain populations that are especially well-suited for further breeding. Although you need to be careful about that conclusion as long as the observed differences between populations have not been reproduced under more controlled conditions.
- However, with the conclusions concerning the photosynthetic parameters and their relationship to the observed survival and overwintering rates I am less convinced. This all seems rather speculative to me. For instance, it looks as if plant performance in 2012, i.e. plant height, stem diameter and number of tillers, were the best predictors of plant survival. Could it be that bigger plants are simply more tolerant to the environmental conditions? For the other parameters, you did observe differences between individuals that survived into 2013 and those that did not. However, in the following season, i.e. 2013, many of these parameters changed into the other direction compared to the previous year, e.g., in the year 2012 the plants that survived into 2013 had a lower stomatal conductance than the ones that were going to die. However, in the following year, the very same plants displayed stomatal conductances that were comparable to the ones of the dying plants in the previous year. How do you interpret this? Could it be, that the high precipitation rates in July 2013, just shortly before you did your measurements, affected the latter?
Methods
- Thank you again for providing all the weather data concerning the performance of your experiment.
- Also, many thanks that you explained the experimental design in more detail. Unfortunately, I still had some difficulties to follow. I think, you could just write this more clearly, e.g., from l.431 on: “For each M. lutarioriparius population 784 individual seedlings were distributed across the 41 plots in such a way that each plot contained (approximately) 19 (?) individuals of each population (i.e. 784 seedlings per plot), adding up to a total of 32,144 individuals that were transplanted to the experimental field in June 2011. Within each plot, the individuals were arranged in a completely randomised design (?). To improve the survival rate of transplanting…”
- 495: you did not show that data in histograms but in bar plots.
Conclusion
In my opinion the conclusion is too long and also should not repeat the results this explicitly but rather summarise the key findings and give a brief outlook (as you did). Please restructure this part.
Supplementary information
Figure S1: Please make clear how you made those measurements. In the figure caption it is stated that 5 leaves were measured repeatedly. However, in your response to my previous comments you say that the measurements were taken from 2 plants repeatedly. Please clarify. Also, the word “mean”, I suppose, is missing before “±SE”.
Figure S4. Please invert the x-axis in such a way that you start at -30 days before conducting the experiment and end up on the right side at -3. The day at which you took the measurements would then correspond to day 0. That would make the figure much more intuitive in my opinion.

Reviewer 2 Report
I'm afraid that the revision of the manuscript has not improved the English style significantly and as a consequence it is very difficult to follow the text. It is essential that the use of the English is improved as at present there are many statements that are confusing and suggest a lack of understanding by the authors.
Round 2
Reviewer 2 Report
The authors have made considerable improvements in the use of English in this revised manuscript. I still suggest a further careful reading and revision of the English before the manuscript proceeds. I hope the work leads to the selection of some useful stress tolerant genotypes for use in future breeding programs.
Author Response
Please see the attachment

This manuscript is a resubmission of an earlier submission. The following is a list of the peer review reports and author responses from that submission.
Round 1
Reviewer 1 Report
Review of Zhao et al.: „Water use efficiency and stress tolerance of the potential energy crop Miscanthus lutarioriparius grown on the Loess Plateau of China”; submitted to Plants
In their manuscript, Zhao et al. investigate the performance of 41 different populations of the perennial grass Miscanthus lutarioriparius that were originally collected along the Yangtse River after they had been translocated to the Loess Plateau, which is characterized by a drier and colder climate compared to the native habitat, through two consecutive growth seasons. In addition to plant survival and growth parameters, such as plant height or stem diameter, parameters related to photosynthesis and carbon assimilation, such as stomatal conductance and intrinsic or extrinsic water use efficiencies, were determined for a subset of individuals once in each of the two growing seasons. This enabled the authors to compare the physiological parameters between individuals that survived the first growing season and the ones that were doomed to die. In addition, for the individuals that survived through both growing seasons, a comparison of the physiological parameters between the two seasons was carried out. Firstly, considerable differences in the survival rates were observed between the 41 populations, but also between the two growing seasons. Secondly, plants that survived the first year did display higher plant heights, stem diameters and tiller numbers compared to the individuals that did not survive the first year. And thirdly, plants that survived the first year did display a lower stomatal conductance, lower intercellular CO2 and lower transpiration rates, but higher intrinsic water use efficiencies. A cluster analysis across all 155 individuals that survived both years and that was based on the various parameters analysed identified five distinct clusters, out of which the authors select one, which, they suggest, includes the populations with the highest potential to be used for breeding of cold and drought tolerant varieties.
While both the aim and the findings of the manuscript are quite interesting and of relevance with respect to the study’s specified aim, the structure and readability can both be improved considerably. Also, I found it quite difficult to follow the authors’ line of argumentation but also such things as the study design. Here, I would wish for more clarity (see below). There are also quite some issues with the language, concerning both grammar and vocabulary. I would recommend the authors to have it checked by some native speakers in order to increase the readability of the manuscript.
Structure of the manuscript:
Actually, I found it quite difficult to read through the manuscript and I think one reason for that was the arrangement of the results shown. In my opinion, it would make more sense to first show the comparison between the plants that survived 2012 + 2013 and the ones that only survived 2012, followed by a comparison of the performance of the 155 surviving plants between the two growing seasons 2012 and 2013 potentially indicative of an acclimation over time, i.e. Fig. 3 before Fig. 2. To me this appears more logical. This would concern the presentation of both the growth and physiological parameters.
Main concerns/comments:
- My main concern, though, relates to the conclusions being made: in my opinion, the authors should be much more careful in the interpretation of their data, especially regarding the comparison between the physiological data from the two different years, 2012 and 2013. For example, in ll. 234 the authors state: “[…], there was no significant difference between two groups of lutarioriparius individuals. Thus, those individuals surviving in both 2012 and 2013 were capable to maintain high photosynthetic rates when the stomatal conductance decreased for reducing water loss in arid environment.” Firstly, to me this interpretation implies causality where only a correlation has been observed, In order to test for causality, additional experiments would be necessary, e.g. by comparing the differently tolerant individuals under controlled conditions in the green house. This directly relates to my second point: in how far are the water use efficiency and stomatal conductance data comparable between the two years? I wonder whether these traits depend on the current weather/soil moisture conditions experienced by the plants while being measured. I have to confess, I am not an expert in these measurements but I would assume that they do depend on these external factors that exceed the authors’ control, and which would render their interpretation somewhat more difficult. Could you please elaborate on this?
- Related to the previous point, could the authors please provide extensive weather data for the study site and for the duration of the study? This would be crucial to better interpret the findings, since in the current form it is not clear what kind of drought conditions the plants experienced, for example. Less harsh conditions in the second year could also be an explanation for the generally higher survival rates that have been observed, instead of a putative acclimation of the surviving individuals as suggested by the authors.
- The differences in plant survival between the various populations look very promising concerning the selection and breeding of more tolerant varieties. However, as stated above, I would wish for a confirmation of these differences under controlled green house conditions that mimic the conditions observed at the experimental site during the sturdy period. Is that something that could be done/has been done or is planned to be done in the near future?
- Concerning the study design: as far as I understood, the 41 populations were grown in 41 separate plots. Is that correct? And if so, how do you exclude position effects?
- Do you have data on the relationship between the performance of the populations at the study site and their native habitat, where they have been collected?
Results:
Please revise this section extensively and try to be clearer about it. To me, the presentation of the results and especially the comparison between surviving and dead plants on the one hand and the two growing seasons 2012 and 2013 for the surviving plants on the other hand (Figures 2, 3, 5 and 6) was really difficult to follow (see suggestion above).
- In general, I would wish for some more detail in the figure captions (e.g. how many individuals are represented by each bar).
- What is actually shown in Figure 4? Is this a measurement of a single plant that was taken on a single day in one of the two years? Is this supposed to be representative for the conditions during the 2 weeks of measurements in 2012 and 2013?
- Table 1: what measures are shown in the table? I suppose r values? Please also specify here but also in the methods section which kind of correlation analysis you used.
- Figure 7: I like the approach of doing a cluster analysis (Fig. 7). I was wondering, though, whether individuals originating from the same population cluster together? Could you please indicate in the figure or in the supplementals which population would be in which cluster.
- In the caption of Fig. 3 there is a mistake. The black bars are supposed to represent individuals that survived only in 2012, and not in 2013 as indicated in the manuscript.
- Could you please provide all data (growth and physiological parameters) for the 41 populations in the supplementals, either as a formatted table or as a separate excel sheet?
Methods:
- Please explain in more detail and clarity the design of your study. This relates especially to the arrangement of the plots? As far as I understood the 41 populations were grown in 41 distinct plots (see my point above). Or was there some kind randomized block design?
- Please revise Fig. 8. In my version there was no red square and also half of the dots were actually yellow (not mentioned in the caption).
Discussion:
Please phrase your conclusions more carefully concerning causation (see my point above). Otherwise, I would be happy to review the discussion in more detail once the requested information on the comparability of the physiological measurements across different years and the respective weather data have been provided.
Reviewer 2 Report
This manuscript describes the measurement of a number of adaptation and yield traits measured on 41 populations of Miscanthus lutarioriparius collected from central China and grown a cold and arid climate to impose serious stress on the plants. A total of 20 individuals with 'outstanding' growth traits and strong resistance to cold and drought were identified. The authors anticipate that these will be used to develop cold and drought tolerant varieties. This is clearly an important exercise but its shortcoming is that it is just reports the first steps of the process. I would like to see the paper showing much more originality and clearly demonstrating how the work will carry forward. At present the work is pretty mundane and tells us very little that is new and exciting.
Unfortunately my reading of the paper was spoilt by the very poor standard of English language and style. Tenses are frequently wrong. Plurals and singulars are often confused. It needs a very thorough revision to resolve this problem. It is also littered with very subjective words like 'excellent', 'always', 'imperative' and 'outstanding'.
The Introduction should finish with the hypotheses, not vague statements about what you intend to do, and certainly not to 'discuss what characters could do' as at present.
The results are confusing in some cases. What does the word 'stipe' mean? In figure 3 the term 'dead' is confusing, explain this more clearly. What is the point of figure 4? When were A and g for 155 individuals measured? It must have taken quite some time. Why are the WUE measure apposing and conflicting? What is the difference between Figures 5 and 6? What is the point of the cluster analysis? Is it described in the Methods?
The discussion is rambling and does not really place the work in the context of other published work. It is largely a restatement of results. The discussion should address the hypotheses.
The conclusions could be much briefer.
Reviewer 3 Report
The manuscript is in good shape for publication, I only detected a minor misspelling on line 94 (HG instead of GH). I suggest the authors should try to reduce to the conclusions, in the current shape, they seem to be a summary of the main results found in the study instead of a conclusion per se. I would also suggest the authors prepare a larger Figure 7 as in its current form it is very difficult for the reader to notice the colors and scale inserted. I would also suggest the authors consider the "standard deviations (error bars)" statement on the captions of Figure 2, Figure 3, Figure 4, Figure 5, and Figure 6, as for me, "error bars" may mislead the readers who may be confusing it with "standard error" and what the authors are in fact presenting are the standard deviations.
suggestions for the authors of the manuscript(attached):
INTRODUCTION
- Line 41. Space before the bracket. Please check for this mistake as it is repeated frequently in the entire document.
- Line 49. "have been" instead of "has been", eliminate "the"
- Line 52-53. Avoid the excessive use of "the", please considerer modifying this sentence to "Miscanthus, as a promising bioenergy crop, can not only support bioenergy production but also resolve ecological problems". Revise the entire document for this suggestion
- Line 59-61. The sentence is confusing and contains unnecessary replication of information. Consider changing it to "Miscanthus lutarioriparius has distributed natively in the middle and lower reaches of the Yangtze River [21] with annual rainfall ranging between 1000-1400mm"
- Line 69-70. Please consider changing this sentence to:
It was defined as the carbon assimilated per unit of water transpired, which is a particularly significant factor for energy crops under drought stress conditions"
- Line 73-74. Explain what you mean by "regulated synthetically"...
Regulated by the stoma and photosynthetic rate? Why not "by the transpiration and photosynthetic rate" or "by the stoma and the chloroplasts"?
- Line 80-83. Please consider changing to:
"Simultaneously, it has been proposed that low stomatal conductance and high intercellular CO2 concentration could mitigate drought stress [38]; however, a low stomatal conductance for reducing water loss results in a low CO2 concentration in the leaf, which causes a decrease in photosynthetic rate [36, 39]."
- Line 83-86. Consider changing it to:
"The decline in photosynthetic rate and the increase in WUE under drought stress were in favor of the plant survival but were unable to contribute to increasing the biomass yield [8]"
- Please, explain what you mean by "It implied..." in line 85.
- Line 94: HG?
- Line 93-97. Please consider:
"(1) to identify the populations of M. lutarioriparius capable of surviving in the GH; (2) to discuss what character of M. lutarioriparius could be related to the growth and survival in a drought environment; (3) to discuss what character of M. lutarioriparius could be an adaptable trait capable of being improved so that the plants can survive in a prolonged growing season in a drought and cold environment."
Why in the second objective is the drought environment included but the cold environment omitted?
RESULTS
- Figure 1. Use "2012" and "2013" instead of "The 2021 growing season"
- Figure 1. Y axis, use "Survival rate (%)"
- Line 116. Figure 2 shows significant differences only for tiller number between growing seasons
- Line 122. "stripe"?
- Figure 3 (Line 130). Explain how the black strips are for individuals that survive only in 2013 but at the same time is for "dead" individuals...seems contradictory
- Line 137. I see in Figure 4 that this was between 10 and 17 hours
- Line 142-149. Do not repeat data already stated on Tables or Figures...Please, double-check that statement on "Additionally, the photosynthetic rate, stomatal conductance, intercellular CO2 concentration and transpiration rate presented a higher value in those individuals surviving only in 2012 than those surviving in both 2012 and 2013 and the increase except for in photosynthetic rate presented a significant difference" because Figure 6 do not coincide with such statement
- Line 157-160. Do not repeat data from Tables or Figures
- Line 170-173. It seems the sentence is incomplete as the word "although" implies the authors are going to contrast some facts
- Line 177-179. The symbols (A, E, g, etc.) for gas exchange parameters should be also included in Figures 4, 5 and 6
- Line 181. Which were the criteria for the classification into five groups?
- Figure 7. Use a larger Figure so the reader may observe clearer; the color scale should be easily observed too
DISCUSSION
- Line 199-200. Consider changing it to:
- "In recent years, reports indicate that in the Loess Plateau there were 40 frost days and 140 ice days [41]."
- Line 210-217. Try to use the information reported in the literature to support your results, in addition, re-phrase the paragraph as in its current form is a bit confusing.
- Line 206-207. I agree that the low temperature could have killed the plants, however, the authors are presenting the mean air temperature (6.7-9.2 C) for the Loess Plateau; I suggest the authors should also provide the absolute minimum temperature so that one can make inferences as to what is the threshold this species can tolerate
- Line 209. "achieve" would be a better word than "conquer"
- Line 210-211. If the statement in this sentence is based on your results, why is it necessary to include a reference "[14]"?
- Line 214-215. Photosynthesis is vital to plant development, biomass production, and seed yield and it is extremely sensitive to drought stress
- Line 224-225. Please, re-phrase this sentence
- Line 228. Do not refer to tables or figures in the Discussion section
- Line 239-280. this section of the discussion needs more work as it is confusing; the ideas the authors state are fractionated and in many instances, the authors do not try to link them with the results they obtained. For example, line 254-256 may be changed to stress the results obtained in view of the cited literature:
- "It was reported that M. lutarioriparius had rich genetic variation and strong adaptability to the semiarid environments [24, 26], which was confirmed in our study as even though the majority of the individuals failed to survive in GH, the average survival rate of M. lutarioriparius populations had high variability.
- Line 241. "in our study we observed..."
- Line 243. "on" instead of "of"
- Line 244-245. Please consider:
"It indicated that the tiller number of M. lutarioriparius could be the growth trait that was limited firstly by the prolonged drought and cold environment."
- Line 305. Do not start a sentence with a number, the figure must be spelled out. Check for this mistake as it is everywhere in the manuscript
- Line 303. After the 1-year adaptation period, the survival rate of 41 M. lutarioriparius populations was counted in June 2012 and 2013
- Line 274-276. Here the authors describe the traits for the 4th group, but we cannot read the traits for all the other groups
- Line 279. groups or group?
MATERIALS AND METHODS
- Line 283. The experimental site was located in the...
- Line 286-288. Please, revise the sentence as it is confusing...is 40 days with maximum temperature below 0 C but at the same time, you have 140 days with temperatures below 0 C?
- Line 290. "Forty one"
- Line 300-301. Rephrase the sentence as it cannot be understood
- Line 304. Six hundred and thirty-four out of 1366 individuals that survived ...
- Line 305. Do not start a sentence with a number, the figure must be spelled out. Check for this mistake as it is everywhere in the manuscript
- Line 321. How many plants from each population were sampled for gas exchange parameters measurements? Were all the 41 populations sampled the same day?
- Line 335. upper script
CONSLUSIONS
- The conclusions section is too large; I suggest the authors should try to reduce it

Round 2
Reviewer 2 Report
I'm afraid that I still have great difficulty in following the English language in this revised manuscript. Furthermore, I am still not convinced that the scientific content justifies publication in this journal.